# Is the Insect Cuticle the only Entry Gate for Fungal Infection? Insights into Alternative Modes of Action of Entomopathogenic Fungi

**DOI:** 10.3390/jof5020033

**Published:** 2019-04-16

**Authors:** M. Constanza Mannino, Carla Huarte-Bonnet, Belén Davyt-Colo, Nicolás Pedrini

**Affiliations:** Instituto de Investigaciones Bioquímicas de La Plata (CONICET CCT La Plata-UNLP), Universidad Nacional de La Plata, calles 60 y 120, 1900 La Plata, Argentina; constanza.mannino@gmail.com (M.C.M.); carlahb@hotmail.es (C.H.-B.); bdavyt@gmail.com (B.D.-C.)

**Keywords:** *Beauveria bassiana*, *Metarhizium* spp., oral infection, per os entry, bacterial-like toxins

## Abstract

Entomopathogenic fungi are the only insect pathogens able to infect their host by adhesion to the surface and penetration through the cuticle. Although the possibility of fungal infection per os was described almost a century ago, there is an information gap of several decades regarding this topic, which was poorly explored due to the continuous elucidation of cuticular infection processes that lead to insect death by mycosis. Recently, with the advent of next-generation sequencing technologies, the genomes of the main entomopathogenic fungi became available, and many fungal genes potentially useful for oral infection were described. Among the entomopathogenic Hypocreales that have been sequenced, *Beauveria bassiana* (Balsamo-Crivelli) Vuillemin (Cordycipitaceae) is the main candidate to explore this pathway since it has a major number of shared genes with other non-fungal pathogens that infect orally, such as *Bacillus thuringiensis* Berliner (Bacillales: Bacillaceae). This finding gives *B. bassiana* a potential advantage over other entomopathogenic fungi: the possibility to infect through both routes, oral and cuticular. In this review, we explore all known entry gates for entomopathogenic fungi, with emphasis on the infection per os. We also set out the fungal infection process in a more integral approach, as a need to exploit its full potential for insect control, considering all of its virulence factors and the conditions needed to improve its virulence against insect that might offer some resistance to the common infection through the cuticle.

## 1. Introduction

Over the last 400 million years, fungi and insects have coexisted, developing interactions in different ways [1,2]. Among them, pathogenicity is a characteristic that has evolved in the major fungal families; becoming the most abundant natural pathogens found in arthropod populations, mostly the hypocrealean *Beauveria bassiana s.l.* and *Metarhizium anisopliae s.l.* [3,4]. The main characteristic of entomopathogenic fungi, unlike other insect pathogens such as bacteria or viruses, is the ability to infect their hosts by penetrating through the cuticle without the need to be ingested. Thus, they have a great potential for controlling sucking insects, which are either agriculture pests (e.g., aphids, leafhoppers, stink bugs, thrips) or disease vectors (e.g., mosquitoes, kissing bugs, tse tse flies). This unique attribute, however, usually represents an obstacle to develop efficient mycoinsecticides due to the slow action (speed of kill) related to their life cycle. Commonly, the first step in the fungal infection process is the conidial adhesion to the insect cuticle by nonspecific interactions, followed by conidial germination and, in many but not all instances, the formation of appressoria (turgor pressure cells) that breach the cuticle [5]. Once inside the hemocoel, fungal cells acquire yeast-like forms called hyphal bodies that invade the host throughout a sequential process, ending in insect mummification. These cells can also secrete small toxic molecules (secondary metabolites) that serve as immunosuppressive compounds, facilitating fungal infection [5]. 

Infection by ingestion is a rare route for entomopathogenic fungi, but quite common for other pathogens such as protozoa, bacteria, and viruses, which display a series of virulence factors that allow them to be ingested and thus infect from the oral cavity and/or intestinal tract [6]. A distinction ought to be made in these cases: oral infection is commonly used to define colonization and mycosis through ingestion, but not making a difference whether the infection process occurs in the buccal cavity or the intestinal tract [6]. At the molecular level, little information is available regarding the oral infection progress by fungi although the whole genome sequencing of *B. bassiana* [7] and *Metarhizium* spp. [8,9] has allowed the identification of some candidate genes. More recently, some metabolic pathways supposedly involved in oral infections by the entomopathogenic fungus *B. bassiana* became available in the context of insect-pathogen coevolutionary studies [10]. Thus, the ability to infect through both cuticular and oral routes (Figure 1) would make *B. bassiana* (among fungal pathogens) a harder challenge to the insect’s immune system, thereby a good candidate to optimize sustainable and ecofriendly alternatives to chemical pesticides. In the following sections, we present information about all known routes of entry for hypocrealean entomopathogenic fungi, with emphasis in the least explored, per os, insect infection route. 

## 2. Cuticular Infection Route

The first contact that an entomopathogenic fungus makes with its host is achieved by nonspecific hydrophobic and electrostatic interactions between conidia and the insect cuticle; some species even produce a mucus substance that helps fungal adhesion [11]. As was reviewed by Butt et al. [12], the main actors at this point are proteins such as adhesin-like (*Mad1, Mad2*) and hydrophobins or hydrophobin-like proteins (*Hyd1*, *Hyd2*, *Hyd3*, *SsgA*, *Cwp10*) [12] (Figure 1). Then, cuticle degradation starts at the most external layer (named epicuticle), which is mainly composed of very long chain hydrocarbons [13]. A suite of cytochrome P450 genes (*CYP*) involved in insect hydrocarbon degradation was described long ago for *B. bassiana* [14], and some of these genes were functionally characterized in *B. bassiana* (*CYP52X1*) [15] and *M. robertsii* (*CYP52X2*) [16]. Immediately after passing through the epicuticle, the procuticle, composed by chitin and proteins, must be also crossed to reach the hemolymph. Entomopathogenic fungi have evolved a variety of degradative chitinases and proteases—chitinases *Chit1*, *Chit2*, *Chit3*, *Chit4*, and subtilisin, trypsin, and cysteine proteases (*Pr1*, *Pr2*, *Pr4*) [12]—that act in combination with mechanical pressing to rupture and pass through the different procuticular layers [17,18]. Furthermore, fungal chitin deacetylases catalyze partial deacetylation of chitin, producing chitosan, a glucosamine polymer that is then hydrolyzed by chitosanases [19,20]. Both enzyme families contribute cooperatively to chitin degradation and facilitate cuticle penetration by fungi. There are also stress-related genes—heat shock protein family and mitogen-activated protein kinase—that aid the fungi in the invasion process and prevent from damage from the insects´ immune system [12].

Once the cuticle is breached, the pathogen reaches the hemolymph, where it needs to adapt to the new environment while it continues to fight against insect immune system [5] (Figure 1). The fungal development inside the insect cavity usually occurs as hyphal bodies, which have a high ratio of surface area-to-volume, improving nutrient absorption. In order to overcome host defenses, but also to tolerate the high osmotic pressure inside the hemocoel, fungi express some genes acting as osmosensors (*Mos1*) or encoding for a defensive coating protein (*Mcl1*), which help hyphal bodies to resist the hard environment found in the hemolymph. Furthermore, antioxidant systems are triggered to mitigate reactive oxygen and nitrogen species (ROS and RNS) released by the host [5,12,21]. Efficient nutrient absorption gives entomopathogenic fungi the possibility for the rapid and massive increase of fungal biomass inside the host, by expressing some genes involved in nutrient uptake such as *Nrr1*, *Crr1*, *Mest1*, *Atm1*, and *Gat*, among others [12]. Secondary metabolites have the leading role in this scenario, since this group of compounds is crucial for entomopathogenic fungi survival and their interaction with other organisms [22]. *Beauveria bassiana* produces secondary metabolites acting as immunosuppressants, facilitating infection, such as beauvericin (Bea), bassianolide (Bsl), oosporein (Op), tenellin (Ten), bassiantin, and beauverolides [5], whereas *Metarhizium* spp. produce mainly destruxins (Dtx) [5,12] (Figure 1). Secondary metabolites pose antibacterial and antifungal properties, preventing the growth of opportunistic saprophytic bacteria and fungi [23]. 

## 3. Alternative Infection Routes for Entomopathogenic Fungi

Insect infection by microbial ingestion is the rule when the pathogen is a virus, bacteria, or protozoa; however, it has also been proposed that entomopathogenic fungi can use oral and respiratory routes as an alternative to the cuticle penetration for entrance to the host [6,10,24,25,26]. These alternatives can signify an opportunity to increase effectiveness against fungal-resistant arthropods that embed their cuticle with antifungal compounds [27,28,29]. The first reports on alternative infection routes for entomopathogenic fungi were published decades ago [30,31,32]. After several years without novel information, in the last decade, several cues about the molecular mechanisms underlying these infection routes started to be studied. This was possible by the advent of next-generation technologies, which produced vast genomic and transcriptomic information from both fungus and arthropods [7,33,34]. 

### 3.1. Oral Infection Route in Terrestrial Insects

Studies on the possibilities of oral/intestinal infection by entomopathogenic fungi have been of interest for many years, and yet, there is much to elucidate on how these routes work. Almost 100 years ago, Peirson [32] hypothesized that *B. bassiana* could invade the oral cavity of pine weevils. From the mid-1940s to the mid-1980s, a discontinued series of research started to gaze into the alternative ways that a fungal pathogen had to colonize its host; the presence of *M. anisopliae* hyphae around the implantation of the mandibles and esophagus of *Ephestia kuhniella* Zeller (Lepidoptera: Pyralidae) [35,36] and recovery of non-germinated spores of *M. anisopliae* from the gut of *Oryctes* larvae were documented [37]. Veen [31] fed second instar larvae of the desert locust *Schistocerca gregaria* Forskål (Orthoptera: Acrididae) with *M. anisopliae-*infected leaves to test oral infection, disinfecting the insect heads to prevent cuticular infection. This research showed that fungal hyphae were present only in the maxillary palps and head, but not in other parts of the body [31]. 

A very thorough study to evaluate the oral infection of pine weevil, *Hylobius pales* Herbst (Coleoptera: Curculionidae), was reported by Schabel [6]. The insects were force-fed with *M. anisopliae* conidia, then conidia viability was determined, and histological analyses of the insect gut were carried out. High mortality was achieved, and hyphae growth was observed in all portions of the digestive tube; however, no clear evidence of germination in the gut was observed, suggesting that fungal conidia invade through the beetle mouthparts [6]. Furthermore, *S. gregaria* fed with *M. anisopliae* conidia produced similar results, reinforcing the hypothesis that rather than penetrating the gut, which is protected by its microbiota, fungal conidia adhere to, germinate on, and penetrate through the buccal apparatus to kill the insects [38,39]. The Colorado potato beetle, *Leptinotarsa decemlineata* Say (Coleoptera: Chrysomelidae), was also fed with fungal conidia, but in this case, *B. bassiana* was used, as reported by Allee et al. [40]; germinated conidia were found in the gut with little to no impact of the gut microflora, although the authors also reported that hyphae penetrated from the outside from fungi that had germinated in the integument due to frass contamination [40]. In contrast, fungus-infected *Bombyx mori* exhibited midgut cells dissolved by fungal metabolites and hyphae growing from the midgut out [41]. A more recent study on *Sitophilus granarius* Linnaeus (Coleoptera: Curculionidae) fed with a mixture of conidia and diatomaceous earth, where insects cuticles were disinfected to prevent cuticular breaching, showed that ingested conidia of *B. bassiana* and *M. anisopliae* were able to infect the digestive tube and kill the beetle [25]. Nevertheless, this study lacked histological information to support the proposed hypothesis strongly. 

This suite of studies on terrestrial insects showed that the mechanisms that ingested spores use to kill the host are not very clear. In some cases, it appears that fungal spores would adhere to parts of the buccal cavity more than the digestive tract, since no germination of conidia has been detected in the gut. Further information and careful experimental design are needed to shed some light on physiological and molecular changes when entomopathogenic fungal spores are ingested. 

### 3.2. Oral Infection Routes in Aquatic Insects: The Particular Case of Mosquito Larvae

Mosquito larvae, which grow in an aquatic environment, seem to offer some choices for infection by entomopathogenic fungi adapted to terrestrial hosts. However, the species have not evolved to interact, and although fungi retain pre-formed pathogenic determinants that mediate host mortality, they do not recognize and colonize their host as true aquatic fungal pathogens do [5]. The conidia of *M. anisopliae* float in water since they have a hydrophobic surface. Thus, when the larvae open their perispiracular valves for air intake, conidia reach the surface of the insect and attach at the syphon tip; then, hyphae grow into the trachea and can kill the insect by suffocation [30]. If the same conidia are offered with a non-ionic detergent that deposits the spores at the bottom of the container, larvae can eat them. Conidia may be found in the gut and kill the insect by toxin secretion, but do not invade the rest of the host [30]. 

Dietary stress is thought to have a big impact in this last route of entry since conidia represent indigestible material for larvae and occupy the digestive tract, limiting the host’s access to ingested nutrients [42]. Reports on entomopathogenic fungi killing mosquito larvae agree that the ingestion of conidia takes place, but how the infection proceeds after the entry point differs depending on the host and the pathogen. *Culex* spp. and *Anopheles* spp. do not harm conidial viability upon passage through the digestive tract [43], but *M. anisopliae* conidia within the gut of *S. gregaria* showed fungitoxicity dependent on the gut flora [39]. For *Aedes aegypti* Linnaeus (Diptera: Culicidae), the two primary infection sites were the head and the anal region when fed with *B. bassiana* conidia; nevertheless, the most preferred site for fungal development was the larval gut [44]. Regarding mechanisms involved in conidial ingestion by mosquito larvae, Butt et al. [45] demonstrated that although ingested conidia fail to germinate and are expelled in fecal pellets, insect mortality appears to be linked to autolysis triggered by caspases; i.e., enzymes with protease activity involved in apoptotic processes. The pathway is regulated by Hsp70 and inhibited, in infected larvae, by protease inhibitors.

A very peculiar entomopathogenic fungus, *Culicinomyces* spp., that infects aquatic larvae of culicid dipterans has the ability to adhere and to colonize the insect by adhesion to the anal papillae and not to the cuticle [46,47]. To a lesser extent, this fungal species can also invade the host through the intestinal cavity, but curiously, it does not attach to the cuticle [46]. Rodrigues et al. [48] reported the insecticidal activity of several isolates belonging to the species *Culicinomyces clavisporus* Couch, Romney & Rao (Hypocreales: Cordycipitaceae), and *Culicinomyces bisporalis* Sigler, Frances & Panter (Hypocreales: Cordycipitaceae), on larvae, eggs, and adults of *A. aegypti*, including transstadial transmission. They also found that after repeated serial repassages through *A. aegypti* larvae, *C. clavisporus* killed faster and with a lower dose, although it was not clear whether the repassages improved the virulence or might have restored some of the original insecticidal activity [49].

## 4. What Is Known at the Molecular Level? Candidates and Shared Pathways

The acquisition of massive data through genomics and transcriptomics, mostly in the last decade, uncovered much valuable information for starting to understand the complex molecular mechanisms underlying these host-pathogen relationships. Pathogens that traditionally invade the host through ingestion such as the entomopathogenic bacterium, *Bacillus thuringiensis*, include in their genomes a battery of virulence factors that allow them to invade the insect and kill it from the gut [50]. By mining into the available genomes of entomopathogenic fungi, some of them exhibit a repertoire of homologous genes that would allow fungi to possess oral toxicity [7,8,9] (Table 1). *B. bassiana* has at least 13 heat-labile bacteria-like enterotoxins compared to six in *M. robertsii* and one or none in the rest of the entomopathogenic fungi; additionally, *B. bassiana* also has eight Cry-like delta enterotoxins and three bacteria-like zeta toxins proteins, whereas other entomopathogenic fungi have one or none, suggesting that *B. bassiana* would have a greater oral toxicity than other entomopathogenic fungi [7].

By genetic manipulation, the vegetative insecticidal protein Vip3A from *B. thuringiensis* was integrated into the *B. bassiana* genome to target the larvae of the oriental leafworm moth *Spodoptera litura* Fabricius (Lepidoptera: Noctuidae) through conidial ingestion; the authors observed that the engineered strain was significantly more virulent than the wild type, but virulence was not modified when infection occurred via the cuticular route [51]. In a coevolution experiment context, the feeding of *B. bassiana* to red flour beetle *Tribolium castaneum* Herbst (Coleoptera: Tenebrionidae) resulted in infections, but showed no differences in the activation of the prophenol oxidase pathway, which is activated when the fungal pathogen penetrates through the insect cuticle [10]. Evidence of the oral infection pathway in *T. castaneum* may be the reason why it was difficult to explain evolved and acquired resistance to *B. bassiana* since it would be able to infect through both cuticular and oral routes [10]. It was suggested that *T. castaneum* cross-resistance coevolved between *B. thuringiensis* and *B. bassiana* [24]. Thus, since the *B. bassiana* genome contains Cry-like toxins similar to those found in *B. thuringiensis*, both pathogens may share mechanisms of infection such as oral toxicity [24].

## 5. Conclusions and Perspectives

Biocontrol of insect pest and vectors has been a safe and eco-friendly alternative to chemical insecticides around the world [3]. Entomopathogenic fungi are part of integrated pest control programs and vector management, and their more widely-studied ability to attach and to penetrate through the cuticle is their most attractive feature for use in controlling sucking insects. Nevertheless, the efficacy and effectiveness of the many available commercial products are still considered poor compared to chemicals [52]. For this reason, there is a need to explore the versatility existing in the infection routes of entomopathogenic fungi in order to improve the kill rates these pathogens display. As early as 1921 [32], the possibilities of oral infection by different entomopathogenic fungi started to be evaluated with diverse results. Fungi can infect and colonize at least to some extent when fed to the insects, but this is highly dependent on the specific fungal species and the host. 

Most past studies lacked an optimal experimental design in order to eliminate the possibilities of contamination and infection by cuticle attachment; but successful experiments have demonstrated that even if ingested, *Metarhizium* spp. is not able to germinate in the host gut. Many factors may influence per os infection, among which conidia must be viable and germinate in the gut conditions (i.e., unfavorable pH, presence of digestive enzymes, and microbial gut flora) and must also have sufficient contact time with the gut wall to allow germination and penetration [40]. Host death after conidia ingestion was mostly attributed to infection through the mouthparts or exposed cuticle in the anus region and rapid invasion of the head and tracheae rather than the gut, although there were some cases where fungal toxins or starvation seemed to kill the insect from the gut without actually invading the hemocoel. Nevertheless, *B. bassiana* has potential over other fungi to display greater oral toxicity based on the several genes involved in virulence by oral infection that it shares with bacterial pathogens [24]. This topic promises further advances and novel insights that will allow a better understanding of oral infection by entomopathogenic fungi, particularly *B. bassiana*, which seems to be the most suitable candidate to act through both cuticular and oral infection routes.

## Figures and Tables

**Figure 1 jof-05-00033-f001:**
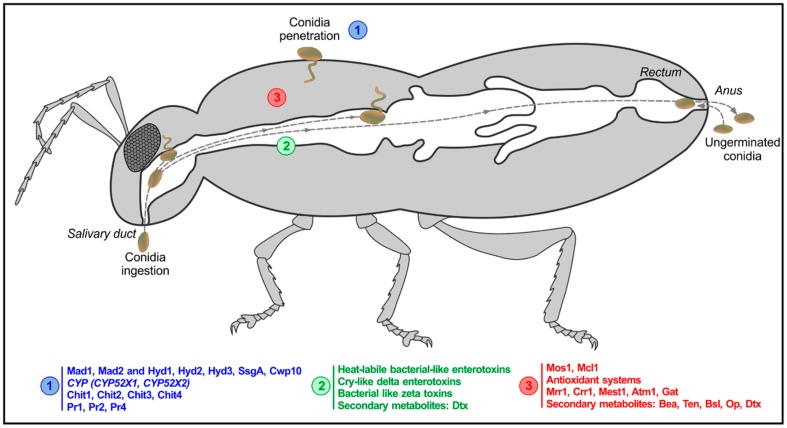
Alternative routes of entry to a host for an entomopathogenic fungus. The fungal genes involved in conidial penetration through the cuticle are shown in blue (1); the fungal genes proposed to participate in oral infection are shown in green (2); and the fungal genes expressed into the hemolymph are shown in red (3).

**Table 1 jof-05-00033-t001:** Entomopathogenic fungal genes potentially involved in oral infection, according to Xiao et al. (2012).

Gene Family	Description	Number of Genes	
		*Beauveria bassiana*	*Metarhizium robertsii*
Heat labile bacterial-like toxins	Bacterial heat labile enterotoxin IIB, A chain (enzymatic) and IIA A	13	6
Cry-like delta enterotoxins	Bacterial delta endotoxin, N-terminal	8	0
Zeta toxins, bacterial-like	Bacterial toxin	3	0

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
