# Peer review of "Is the Insect Cuticle the only Entry Gate for Fungal Infection? Insights into Alternative Modes of Action of Entomopathogenic Fungi"

_jof, 2019, doi:10.3390/jof5020033_

Round 1
Reviewer 1 Report
Dear Authors,
the manuscript is well written I did not find any misspelling nor mistakes to make corrections. I would like to congratulate the authors for such nice manuscript.
This review is very complete because there are not very much information on this subject.
Just one question
Line 166 caspase activity. Could the authors inform what does this enzyme do? I am not aware about this.
Author Response
We appreciate the comments of the reviewer.
Caspases are enzymes with protease activity involved in apoptotic pathways. As mentioned in Butt et al. (2013), caspase activity may be triggered by fungal conidia in infected insects since their results showed high mortality rates are related to caspase activity. They also observed that caspase activity increased in time, compromising more cells to apoptosis.
This definition was included in the revised version.
Reviewer 2 Report
It is a well written "Literature review" that explored known entries gates for entomopathogenic fungi with especial emphasis on the infection per os. It seems to be missing much more information of alternative infection routes particularly with oral infection of B. bassiana since this is what the authors stated on the abstract.
Numeral 3.2 it somehow hanging there. It should be called "Oral infection routes in aquatic insects"??? and definitely a numeral 3.3 needs to be added which will include "oral infection routes in sucking insects" which will be covering the information that the authors are trying to present in this "Note Review".
I am not sure about the title, I think it will be better if the first part of it is omitted, leaving only "Insights into Alternative Models of Action of Entomopathogenic Fungi".
My overall recommendation is Reconsider after major revision.
Author Response
We appreciate the suggestions made by the reviewer.
Regarding the details on alternative routes of infection by B. bassiana, little data is available and was included in subheading 4, since all of them were obtained by molecular tools. These reports suggest that B. bassiana can infect orally, based on the genes activated in the host when conidia are ingested.
The reviewer is right regarding the comment on subheading 3.2. In order to be consistent with the subheading 3.1, it was rewritten as: “Oral infection routes in aquatic insects: A particular case, the mosquito larvae”.
There is no information in literature about oral infection routes in sucking insects; this might be related to the difficulties for deliver conidia into hosts (animals, plants, etc). However, the recent advances in endophytic activity of entomopathogenic fungi could provide interesting information about this issue soon.
The question included in the title intends to give the reader an "at the glance" idea about the aim of the review. At the same time, we believe that referring first to the more usual route of infection is useful to introduce -in the second part of the title- the "alternative" to what process the article is referring to.
Round 2
Reviewer 2 Report
Well written. A few minor points noted on the text

Author Response
All minor corrections were addressed.
Editor review comments:
The manuscript (MS) contains valuable information and will help boosting novel research in the non-cannonical or unusual mechanisms of insect infection (mainly oral but also anal or intestinal ways). The MS is well written and structured. In the initial part, we suggest authors include chitosan metabolism (chitosanases, chitindeacetylases,..) as part of the molecular components for entomopathogenic fungi (EF) to infect insects. There are some publications which support this which complements other genes encoding enzymes (proteases, chitinases, lipid degrading/modifying enzymes) and secondary metabolites (toxins...).
p.7 l.141 "frass contamination". Do authors mean "cross contamination"?
The part of aquatic insects is very interesting but also lacks some references to molecular determinants. The part of genome mining is interesting and well written specially regarding searching for Bt toxin homologues in Bb genome. Finally, the suggestions for future work in the conclusions are sound.
Authors responses:
We appreciate the Editor’s comments.
We have included a paragraph (with the corresponding references) about the activity of chitindeacetylases and chitosanases and their role on fungal penetration through the procuticle.
It is correct the word “frass”. It is referring to a fine powdery refuse, often fecal contamination, produced by the activity of some insects.